# Assessing the Effectiveness of Sustainable Drainage Systems (SuDS): Interventions, Impacts and Challenges

**Sarah Cotterill** [1,*] **and Louise J. Bracken** [2]

1    School of Civil Engineering, University College Dublin, Belfield, Dublin 4, Ireland
2    Department of Geography, Durham University, Durham DH1 3LE, UK; l.j.bracken@dur.ac.uk
*    Correspondence: sarah.cotterill@ucd.ie

**Abstract:** Sustainable drainage systems (SuDS) can be a key tool in the management of extremes of rainfall, due to their capacity to attenuate and treat surface water. Yet, implementation is a complex process, requiring buy-in from multiple stakeholders. Buy-in is often undermined by a lack of practical evidence and monitoring of implemented SuDS. In this paper, we present a collaborative case study between a local authority, university and the UK Environment Agency. This partnership approach enabled the installation of SuDS and monitoring equipment to address surface runoff in the north east of England. Ultrasonic sensors were installed in the drainage network to evaluate the attenuation of surface water. SuDS were installed during an atypically wet spring, followed by a hot and dry summer, providing a range of conditions to assess their performance. Results demonstrate that there was a statistically significant difference in the detected flow level in manholes downstream of the SuDS interventions. Several challenges occurred, from signal obstacles in wireless telecommunication services, to logistical constraints of installing sensors in the drainage network, and issues with the adoption of property level SuDS. These issues require further research. Qualitative support for partnership working was crucial to increase the capacity for delivering SuDS. To ensure the success of future schemes and likelihood of SuDS uptake, partnership working and engaging with communities is vital.

**Keywords:** monitoring; stakeholder engagement; sustainable drainage systems (SuDS)

## 1. Introduction

The management of urban drainage is a critically important challenge and sustainable drainage systems (SuDS) are a key tool in managing extremes of rainfall [1,2]. SuDS consist of a range of technologies and techniques used to drain stormwater and excess surface water in a manner that is more sustainable than conventional solutions [3]. They are based on the philosophy of replicating "as closely as possible the natural, pre-development drainage from a site with installations based on natural hydrological processes which utilise vegetated land surfaces" [4]. Typically, SuDS are configured as a sequence of stormwater practices and technologies that work together to form a management train that is designed to store, attenuate and treat surface water and in so doing, reduce runoff and flooding. SuDS have a secondary role in greening the environment. They are one way in which ecosystems are used to address mounting urban sustainability challenges, giving people access to nature to improve health and wellbeing.

Despite the range of drivers supporting the use of SuDS, their implementation is a complex process and requires buy-in from multiple stakeholders. Buy-in is often undermined by a lack of practical evidence and monitoring of existing schemes to demonstrate their value and best practice.

In this paper, we report one of the first holistic studies of the impact of retrofit SuDS on runoff in an urban environment. Initially, we present the current debates around SuDS, we then outline the study design using a suite of interventions, implemented in collaboration with a range of stakeholders. The results and emerging challenges for assessing the effectiveness and implementation of SuDS are then discussed. The results demonstrate that SuDS can result in a significant reduction in runoff, but this study also highlights the importance of partnership working, and the added value it can deliver in implementing sustainable drainage interventions at the property and the street scale. We also outline the challenges with obtaining high quality data in urban and rural areas.

*1.1. Current Understanding and Evaluation of SuDS*

The use of SuDS in reducing runoff emerged in the UK in the late 1980s. By 1992, the "Scope for Control of Urban Runoff" [5] guidelines were published, providing guidance on a range of technical interventions to control runoff. Further guidance documents were published in 2000 [6], which formalised the term sustainable drainage systems. Since 2000, there have been a number of SuDS regulatory and policy documents published, including DEFRA's "Non-statutory technical standards for sustainable drainage systems" [7]. In some countries, such as Scotland, SuDS have been mandatory in most new developments since 2003 [8].

SuDS are designed to meet specific performance criteria relating to (i) hydraulics, (ii) water quality, (iii) amenity and (iv) biodiversity [9]. SuDS components are generally volume based and designed to manage the interaction between the drainage system and the built environment to facilitate the management of exceedance flows [9]. Additionally, SuDS can provide treatment of the surface water runoff, thereby improving water quality, and in so doing can significantly contribute to the amenity of an area, delivering added value and improving wellbeing.

Papers modelling the impacts of SuDS on runoff are becoming more common (e.g., [10,11]) but there are few studies that actively monitor the performance of deployed SuDS. SuDS monitoring is often qualitative and constrained by a lack of budget. McDonald [12] found that SuDS monitoring and evaluation in Scotland is achieved on an "informal, ad-hoc basis and not at regular intervals as recommend by The SuDS Manual" [12]. When monitoring and evaluation is undertaken, it is often descriptive: ranging from an annual site visit to take photographs, to fairly basic, routine maintenance [12]. There are fewer studies still that quantitatively measure the performance of SuDS, particularly in urban areas and/or retrofit scenarios, due to the challenges in designing and implementing these studies [13–15]. Without monitoring and evaluation, it is unknown whether the SuDS systems are under or over performing.

A database of case studies of blue-green infrastructure (BGI), a term which is often used interchangeably with SuDS, was compiled by Kazmierczak and Carter [16] to highlight the processes that supported their implementation, such as governance and stakeholder relationships, rather than the physical components of the BGI. The case studies included public awareness campaigns, such as The Netherlands "Live with Water" and a case study of retrofit SuDS in Augustenborg in Malmö, Sweden (with similar drivers to the project reported in this paper). The neighbourhood of Augustenborg experienced periods of socio-economic decline, and frequently suffered from floods caused by overflowing drainage systems [16]. SuDS, including retention ponds and green roofs, were installed in this collaborative project involving the city council, a social housing company and the local residents. This scheme resulted in a significant reduction in rainwater runoff and improved the visual amenity of the area. Kazmierczak and Carter [16] identified a number of lessons from these case studies relating to championship: raising awareness within organisations, amongst stakeholders and with the wider public; collaborative working; developing a sound evidence base; and monitoring and evaluation. They found that, in many cases, establishing or adopting BGI was a result of the enthusiasm and commitment of one individual, one organisation or a particular stakeholder partnership i.e., a "champion". They noted the importance of engaging residents in the development and implementation of BGI projects to reduce the likelihood of objections and conflicts [16].

The lack of "appropriate monitoring'" and evaluation of SuDS projects is routinely cited as a barrier to their implementation [13,17]. There are few examples of SuDS demonstration sites and therefore, the evidence base—which could be used to drive further investment in BGI and SuDS—is also lacking. As such, the perceived cost of SuDS often outweighs the understated benefits. Additionally, the ownership of SuDS is problematic and can often be linked with several (separately governed) stakeholders. This can present challenges for the funding of SuDS and the ongoing maintenance of these assets.

### 1.2. The Relationship between SuDS and Health and Wellbeing

In the late 2000s, a shift in research and environmental management occurred that linked nature with wellbeing: society was no longer considered to be a passive beneficiary of nature, but could take action to proactively protect, manage and restore natural ecosystems to address societal challenges [18]. This shift in thinking led to the emergence of a new theme of research and practice around nature-based solutions (NBS), as actors sought possibilities to bring about a more systematic approach to understanding the relationship between nature and society. The ambition of NBS is to develop management solutions that work with ecosystems to address mounting urban sustainability challenges and stimulate economic growth and employment through the green economy [19,20].

NBS can be defined as "actions to protect, sustainably manage and restore natural or modified ecosystems, which address societal challenges (these may include climate change, food and water security or natural disasters) effectively and adaptively, while simultaneously providing human wellbeing and biodiversity benefits" [19] (p. 2). One of the drivers for NBS has been the potential benefits to health and wellbeing. The relationship between health and NBS is complex, but evidence suggests that access to greenspace provided by NBS is important for healthy living and can help to mitigate public health risks associated with urbanisation and climate change [21].

Three routes have been proposed to explain the benefits to health: (i) restoring capabilities; (ii) building capacity by enhanced physical activity, improved fitness and reduced obesity; (iii) mitigation of flooding, air pollution and reduction of the urban heat island effect [22–24]. More recently it has been suggested that the presence, accessibility, proximity and "greenness" of NBS influences the magnitude of the health effects [21]. Furthermore, research is beginning to identify how improvements in water environments can affect a range of socio-economic factors, including demographics [25]. When green residential space is present, this not only improves the amenity value for the existing community, but provides a key driver for migration into an area, which can lead to socio-economic change in the long term [25]. Understanding these impacts is strategically important for prioritising locations for implementing water management improvements [25].

### 1.3. Complexities of Implementing SuDS

Successful implementation of SuDS requires collaboration between public and private stakeholders including local authorities, regulators, engineering consultancies, utilities, academics, central government and NGOs. The multi-sector nature of SuDS projects can be problematic, particularly in relation to the adoption, ownership and maintenance of SuDS. Ownership can often be linked with several, separately governed stakeholders, presenting challenges for the funding of SuDS and their ongoing maintenance. A large-scale questionnaire of practitioners, conducted in 2016 by the Chartered Institution of Water and Environmental Management (CIWEM), "The Big SuDS Survey", echoed this challenge, identifying a need for a single adoption method, coordinated by the local authority [17]. However, there were also concerns over the resources and capacity available within the local authority to support SuDS uptake [17].

Many of the barriers to SuDS are often perceived; based on assumptions or estimates, (e.g., perceived cost, site constraints and land requirements), rather than quantitatively evidenced through monitoring and analysis of deployed schemes. This is often a consequence of the scarcity of demonstration sites, which could provide an evidence base to address such barriers [26]. Thus, a lack

of data collection, and uncertainty around the assumptions being made, may limit the uptake and implementation of further SuDS schemes. Another significant challenge for SuDS is around expertise required for evaluating the quality of SuDS deployments. Three-quarters of respondents to The Big SuDS survey perceived there to be a lack of expertise, capacity and skills in local planning authorities for evaluating and advising on the deployment of quality SuDS schemes [17]. Local authorities often have a lack of resources to monitor SuDS, a lack of capacity to enforce planning conditions, and a wealth of other challenges such as skills shortages which remain to be addressed through "capacity-building programmes" [27]. In a later study, Potter and Vilcan highlight that SuDS design and implementation is often constrained by a range of factors including a lack of legislative backing, a poor planning process and a severe lack of resources in local authorities [28]. They suggest that, whilst an integrated, collaborative and innovative approach is required, there are often significant differences between the attitudes and aspirations of the institutions and the professionals working within them, which affects the capacity to deliver real change [28].

Furthermore, the expertise required goes beyond the relevant policy-making institutions. The process of creating practice involves the selective generation of evidence, which includes "all types of science and social science knowledge generated by a process of research and analysis, either within or without the policy making institution" [29] (p. 208). Recognising expertise from a range of people can, therefore, provide more insight that relying on just one expert. Equally important is the mutual understanding between professionals with different bundles of expertise. Bracken and Oughton demonstrated that professionals make strategic choices about how to implement policy depending on the objectives of a particular project [30]. In this way, experts decide whether evidence is "good enough" to act upon, and may choose to ignore expert advice if it does not help meet project objectives. Innovative behaviour that creates new structures and practices is becoming central to delivering good management of land, water and biodiversity. The complexity of problems, breadth and diversity of evidence, speed of legislative change and complex governance structures mean new groupings of expertise have an important role and that investment of time and resource can have pay-offs for meeting multiple objectives in a range of situations [30]. New institutional practices needed to be developed around the implementation of SuDS, as the technology and guidelines were developed to "get things done". These practices are unique to each situation which takes account of the environmental issue, the policy context, the physical location, the relevant organisations, the individuals involved and how they are brought together at a particular point in time. It has, therefore, taken time for the necessary capacity and skills required to implement and evaluate SuDS.

## 2. Materials and Methods

### 2.1. Conceptual Framework

The purpose of this project was to establish a collaborative process for addressing surface water management concerns in an urban environment and to evaluate both the process itself, and the SuDS interventions delivered in order to inform future SuDS design and delivery. The conceptual framework for the research is presented in Figure 1, which demonstrates the interrelationship between drivers for the development of SuDS in the study area, the collaboration necessary to effectively implement a programme of SuDS project delivery of complex and multifaceted solutions and community engagement.

The drivers for this project included place based investment to improve the quality of housing and wellbeing of the study area, the need to find solutions for ongoing challenges around surface water management and the opportunity to develop and evaluate community based testing of SuDS features. Collaboration was central to this project, both between business (Northumbrian Water (NWL)), environmental regulators (the Environment Agency (EA)), local government (Durham County Council (DCC)) and the local community. Collaboration was supported through previous investment and collaboration in the study area and the Water Hub. Durham University's Energy Institute

and DCC were already working with the local community—and, therefore, had gained necessary participant consent for engagement and interaction—on the European-funded Solid Wall Insulation innovation (SWIi) project, which, alongside delivering solid wall insulation to over 175 homes, involved identifying customer behaviours around smart heating controls and innovative peer-to-peer energy advice [31]. Fuel and water poverty are two of the challenges this community face. The Water Hub was a collaborative partnership set in the North East of England between DCC, Durham University, the EA and NWL, which sought to facilitate open innovation across the water sector [32]. Through a scheme funded by the European Regional Development Fund (ERDF), with match funding from the organisations themselves, the project provided testing sites, research collaborations, grants to small and medium enterprises (SMEs) to support the development of innovations, and delivered business support and networking opportunities to SMEs interested in innovating within the water sector [32].

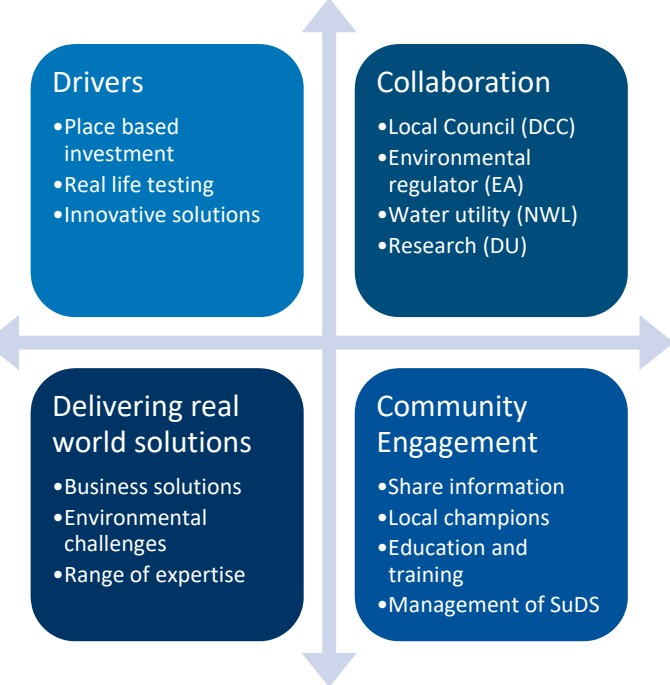

**Figure 1.** The conceptual framework for the study.

The conceptual framework enabled the co-design of a process to bring together investment from the organisations involved in The Water Hub, together with grant funding to deliver and evaluate SuDS interventions in the study area. The process drew on the expertise held across organisations, business, research and the local community.

### 2.2. Study Area

The study took place in the Twizell Burn catchment in the north east of England (Figure 2). This area, which includes Stanley, Quaking Houses and Craghead, has a population of 33,000, and is heavily influenced by historic mining activity [33]. The local river, the River Twizell, is classified as heavily modified under the Water Framework Directive (WFD). It achieves only moderate ecological status [34] as a result of sewage outflows, agricultural pollution, the dewatering of historic mine workings and the mobilisation of sediments from flash flooding incidents [35]. Whilst prior evidence had identified diffuse pollution from industrial estates, the most significant sources of pollution is thought to be from point source discharges direct to the river [35]. In addition to the environmental drivers, there were a number of socio-economic factors driving investment. Seventy percent of working age families in Stanley are in the poverty bracket and almost a third of the 16+ population have no

formal qualifications. Furthermore, one quarter of occupied households claimed housing benefit in 2016, and the proportion of the population living in fuel poverty (13%) is above the national average [33]. Developing SuDS schemes in the study area therefore has both environmental and social drivers, in light of the potential positive impacts on runoff reduction and social wellbeing.

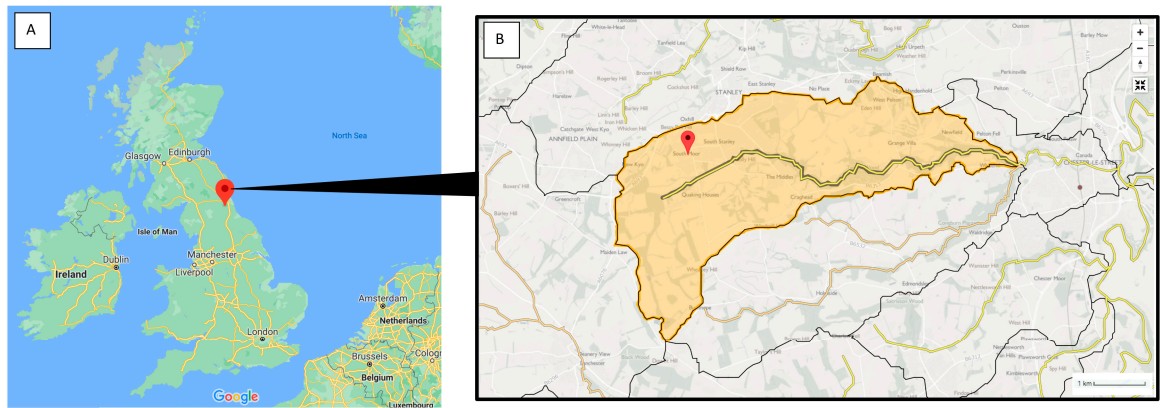

**Figure 2.** Location of the study area (shown by red pin): (**A**) in the United Kingdom [36] and; (**B**) within the Twizell Burn catchment [34].

The study focused on three streets of terraced rows of housing, with no front gardens and small, concreted back yards. There are surface water systems in two of the streets (Elm and Pine). However, these connect to the combined sewer system at Park Road. There are two combined sewer overflows (CSOs) on Park Road, and one relief sewer that diverts excess flows to an adjacent sewer. The lack of pervious surfaces in urban catchments, such as this, has led to an increase in the frequency and magnitude of urban flooding and CSOs [10]. CSO infrastructures are installed in the combined sewer network to limit the volumes conveyed during heavy rainfall. Their function is mainly hydraulic—converting a single inflow into two outflows (to the wastewater treatment plant, and the water body)—to avoid overloading the wastewater treatment plant, and avoid flooding the urban area [37,38]. CSOs can therefore lead to water quality issues and pose concerns for their hydraulic impact on receiving waters, from flood events with short durations and high peak flows [38].

Flooding is common in the study area [35]. Whilst residential and commercial properties have been flooded, the catchment is largely rural and was therefore, at the time of the study, unlikely to be eligible for funding, such as Defra's Flood Defence Grant in Aid (FDGiA) [35]. Therefore, the local authority commissioned an appraisal of SuDS across the catchment, to reduce the impact of stormwater on the combined sewer network, and thereby reduce the number of CSOs discharging to the river [35]. SuDS proposed included raingardens, permeable paving and stormwater planters, following a Surface Water Management study [39]. The Surface Water study included a hydraulic model of the drainage network and topographic survey data, and sought to identify SuDS options which could be installed in six streets in South Moor, alongside complementary interventions, which could be implemented in a wider secondary area of the catchment [35]. This included re-naturalising culverts, enhancing existing wetlands, creating new attenuating wetlands and redesigning CSOs [35]. Due to the space constraints of the study area, smaller footprint SuDS options were identified for retrofitting including permeable paving, stormwater planters, raingardens and street trees. It was noted that options such as swales, attenuation basins and retention ponds, which have a larger land-take, could have a dramatic effect on the capacity and rate at which flows could be attenuated, and therefore these were recommended as complementary options for integration in the wider catchment [35]. The Surface Water Management study was followed by a Green Infrastructure Action Plan [35], which was developed from questionnaires with local people and discussions with relevant environmental managers.

## 2.3. SuDS Interventions

Following the recommendations in the Surface Water Management study [39], a soft market event was held by Durham County Council (DCC), Environment Agency (EA) and The Water Hub ahead of an invitation to tender (ITT) for a programme of work to build and monitor SuDS features [35]. The ITT called for respondents to adopt a collaborative and community-centred design approach, engaging with local residents, SMEs and the existing consultants and partnerships from previous studies (e.g. [35,39]) to design and install: an innovative SuD Tree system on Poplar Street, an affordable rainwater harvesting system for rear yards and a smart monitoring system to quantify the performance of the scheme. A consortium of businesses were selected to deliver a range of interventions, designed to mimic a typical programme of SuDS as frequently implemented by practitioners. The interventions included two SuDS tree pits, four street trees and two rear-yard rainwater attenuation planters (Figure 3). These SuDS features were designed to complement existing rain gardens, porous paving, and feature paving that had been installed in Pine Street, following the South Moor Surface Water Management plan [39].

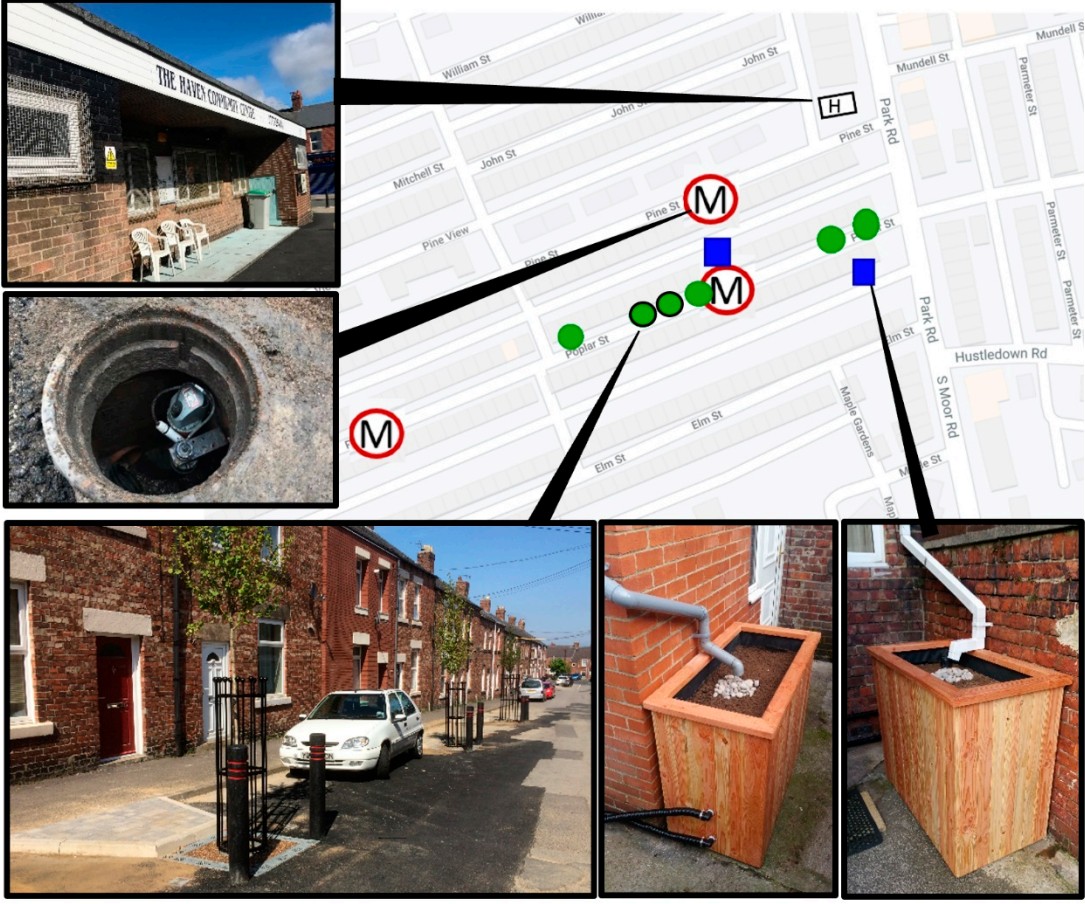

**Figure 3.** The location of three monitoring points (M), two rear yard planters (blue squares), four street trees (green circles) and two sustainable drainage systems (SuDS) tree pits (green circle with black edge) on Pine Street and Poplar Street (not to scale). The dashboard interface was installed in The Haven Community Centre (H).

The two SuDS tree pits were installed in Poplar Street. A 43 m$^3$ trench was excavated to construct the two tree pits. This trench was lined with a porous geotextile membrane and GRN20 plastic reinforcing mesh using a modular cell system (RootSpace, GreenBlue Urban, UK). This load-bearing soil support system was designed for maximum soil and rooting volume in the tree pits (Figure 4A).

Fifty tonnes of Hydrosoil mix was installed, designed to withstand frequent short-term water logging. A SuDS filter drain was then installed and connected to the existing drainage chamber. Inlets were installed to allow surface water to pass through the tree pit (ArborFlow, GreenBlue Urban, UK). A double inlet irrigation and aeration system (RootRain Arborvent, GreenBlue Urban, UK) was installed to maintain long-term soil health (Figure 4B). Trees were planted and supported by a tree grille and frame (ARBPC1507A, GreenBlue Urban, UK), and protected from road traffic by bollards. Four street trees were installed in Poplar Street, either side of the SuDS tree pits, to further break up the hard standing.

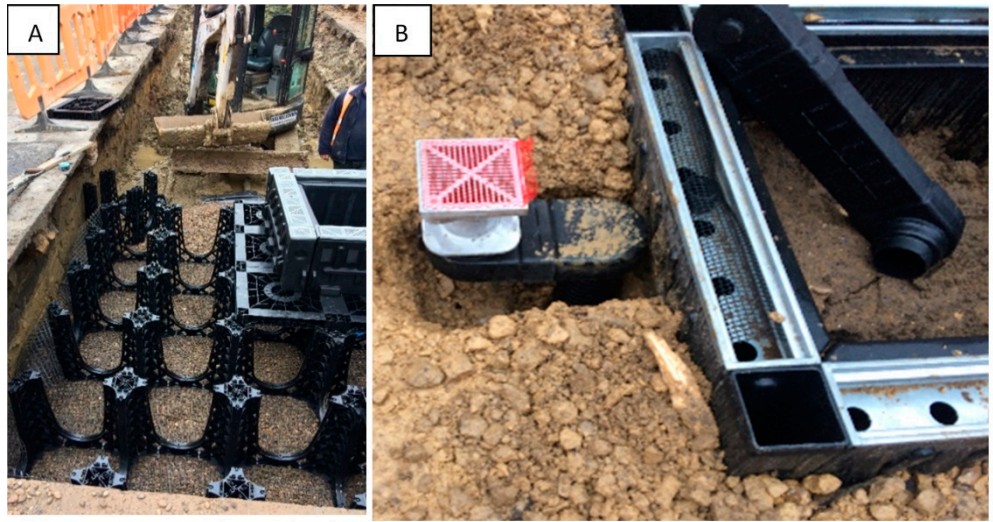

**Figure 4.** Photographs of the installation of two tree pit sustainable drainage systems (SuDS) in County Durham, including: (**A**) modular "RootSpace" cell system and; (**B**) one of the tree pit's aeration vents.

Two rainwater attenuation planters were installed in the rear-yards of Poplar Street to disconnect two downpipes and capture the roof run-off (Figure 5). These planters reduce and slow the flow into the drainage network, reducing the likelihood of the system becoming overloaded. Each planter consisted of a premium quality timber base with a double-wall construction and guaranteed watertight liner (Sudsplanter Ltd., Bradford on Avon, UK). The planter contained layered growing media and filter materials to provide water and nutrients for the plants; and filter and attenuate flow. After filtration and absorption by the plants, any remaining water is collected in a high-volume, high-strength reservoir at the base of the planter, enabling storage for re-use or slow release into the drainage network. Integral flow control ensures a precise discharge rate, whilst a high-level overflow controls water levels during extreme rainfall.

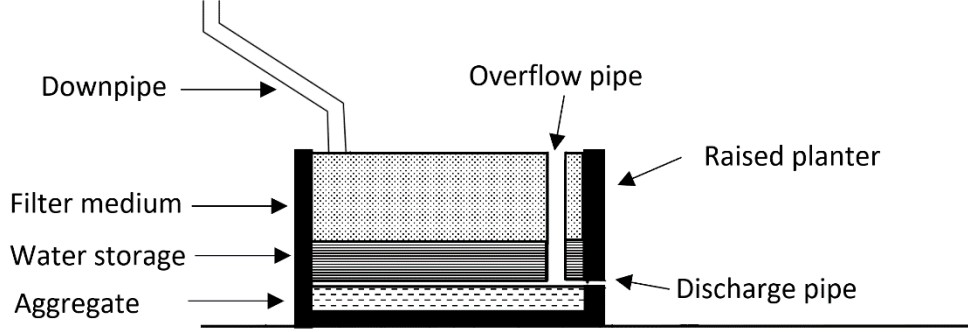

**Figure 5.** Schematic of the rainwater planters installed in two rear-yards on Poplar Street. Each planter contained a filter medium and storage compartment to filter and attenuate roof run-off and reduce flow into the drainage network.

*2.4. Monitoring*

A programme of monitoring was designed to capture any changes in runoff in the drainage system around the interventions at the street level. Flow monitoring solutions were installed at multiple locations within the study area's sewer network, and a telemetered user interface (UI) was developed for the local community centre to enable near real time display of data, with the aim of engaging the local community on the changes that SuDS could deliver.

Establishing the monitoring was not straight forward since the project was undertaken in a real-life setting. A series of consultations with the project team and stakeholders was undertaken to determine the type of monitoring equipment to be used, and the location in which these devices would be placed. Sensors installed in drainage networks must be able to provide continuous recording of data, remote data transmission and be resistant to water and chemicals [37]. Water level measuring devices are most commonly used for these applications [37]. Several options were considered, including v-notch inserts and pressure sensors, area velocity flow meters and ultrasonic level-to-flow meters. Five manholes were identified as priorities in which to install monitoring devices, and a secondary set were identified as potential alternatives. A survey was undertaken of the utility's and local authority's manholes in the study area. An inspection of the proposed locations identified a number of constraints. Some of the proposed locations were not suitable for monitoring in terms of access, space and condition. For example, some of the sewers did not have inspection chambers present, the flow did not always accurately map onto the plan and some of the manhole covers could not be lifted, as they were seized.

Following this initial investigation, seven manholes were surveyed by the civil engineering contractor and engineers from the monitoring team (Figure 6). Three of the manholes (2, 3 and 5) were deemed unsuitable, due to high level inflows and/or inaccessible manhole covers. The remaining four manholes (1, 4, 6 and 7) were evaluated for sensor installation. However, manhole 4 was later removed from consideration due to an unusual high inflow. The three manholes chosen for the study (1, 6 and 7) were located next to SuDS interventions; manhole 1 was downstream of street level raingardens on Pine Street, manhole 6 was upstream of the proposed SuDS tree pit locations on Poplar Street, and manhole 7 was downstream of the proposed tree pits on Poplar Street. All three manholes had suitable chambers to accommodate monitoring equipment on evaluation.

An ultrasonic sensor and bracket were drilled and installed on to the upper wall of each of the three manhole chambers. Each mount point was chosen to provide the most accurate level data achievable in the chambers. A remote terminal unit (RTU) was installed in each of the chambers, although it was acknowledged that this may need to be later surface mounted, due to signal strength. Following this, the micro-PC was configured and installed in the local community centre, connecting to an active internet connection within the building. Finally, the user interface (UI) was configured and programmed to provide a dashboard showing level, calculated flow, system conditions and rainfall information, collected via a rain gauge installed on the roof of the community centre. Level data was recorded by the ultrasonic sensor in each manhole at 5 min intervals, providing up to 288 individual data points per manhole per day.

*2.5. The Roles of Stakeholders*

Initiating deployment of SuDS interventions and monitoring in real life communities is difficult. We therefore developed a novel approach with local stakeholders to support the delivery of the research outline above. The co-developed tender sought to focus on place-based investment, community-based test facilities and innovative solutions to water infrastructure problems. A team of five businesses were brought together to retrofit water attenuation features, install technology to monitor performance and establish an engaged community group to support further investment in green infrastructure. Consultations and planning involved representatives from the water utility, the local authority (DCC), the EA and the team carrying out the civil engineering and groundworks. A programme of community engagement was undertaken alongside the structural work in the project. Due to their involvement in previous projects in the area, engagement was led by DCC. Letters were delivered to residents and

door-to-door visits were made to inform and answer concerns about the impact of the proposed work. "Community champions" were also established to encourage the community to participate in informal drop in events (including gardening and maintenance of SuDS features), and training and education around how to use the digital interface. Qualitative feedback was collected from those involved in all stages of the project, both the project partners in The Water Hub and the community.

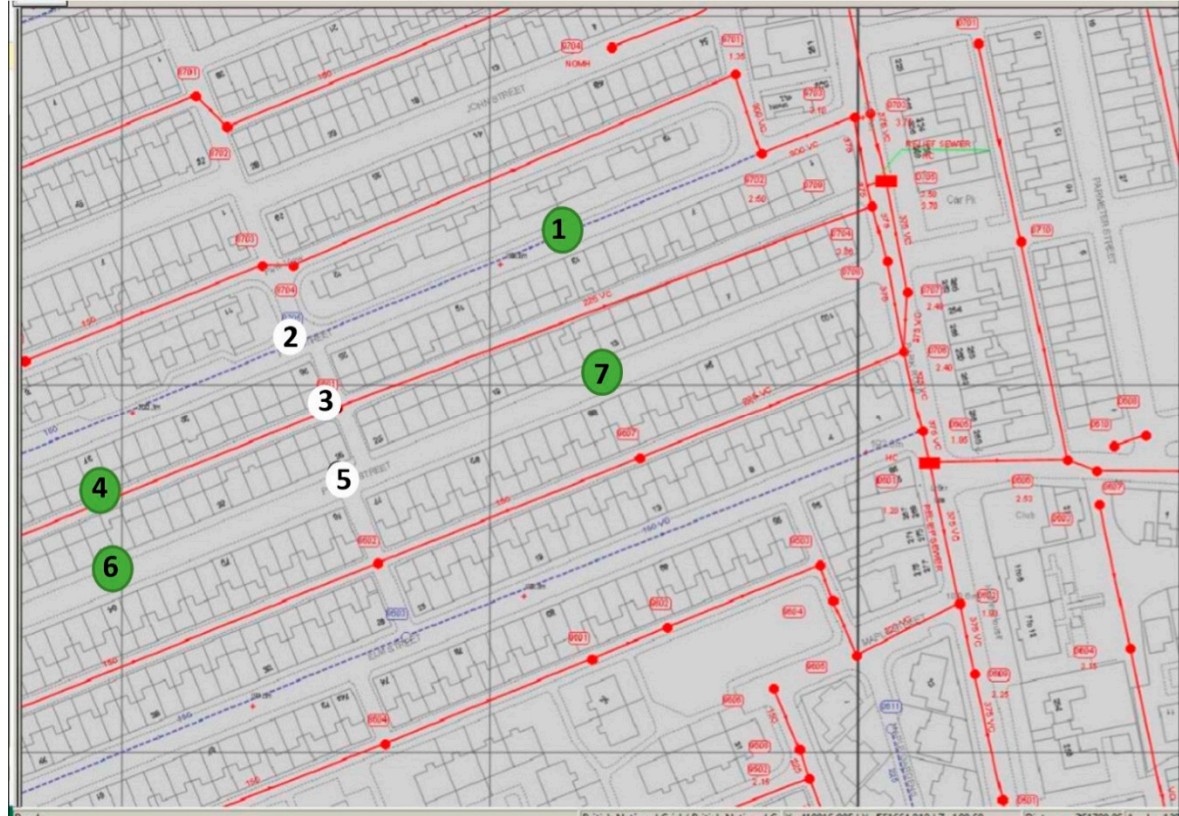

**Figure 6.** Locations for monitoring equipment in South Moor's sewer network following inspection in late February 2018. Green dots show manholes suitable for installation of sensor, white dots show manholes not used in this study. Red lines highlight foul or combined sewers, and blue lines show surface water sewers. The position of private drains and sewers are not shown.

### 2.6. Statistical Analyses

Statistical tests were carried out using IBM SPSS statistics 26 (IBM Corp., New York, NY, USA). Analyses were carried out to determine if there was a statistically significant difference in the flow level (in mm) recorded by the ultrasonic sensor in the manhole before and after the SuDS interventions on Popular Street. A paired-samples t-test was run to determine whether the mean difference between paired observations of flow level in the two manholes was statistically significantly different from zero. The paired-samples t-test assumes there is one continuous dependent variable and one independent variable, which consists of two related groups or matched pairs. Furthermore, it assumes there should be no significant outliers and the distribution of the differences should be approximately normally distributed. The assumption of normality of data is assessed by Shapiro–Wilk's test. The paired samples t-test is robust to violations of normality. However, a non-parametric alternative to the paired-sample t-test, the Wilcoxon signed-rank test, can be used if the data is non-normally distributed. The Wilcoxon signed-rank test assumes that the distribution of differences is symmetrical. If this assumption is violated, the related-samples sign test can be used instead.

## 3. Results

### 3.1. Weather

The planters were installed in March 2018, after a period of above average rainfall (March 2018 = 76.4 mm, Average = 44.3 ± 25.8 mm [39], Figure 7). In the month following installation, rainfall was 70% higher (75 mm) than the mean historic average for April (44.2 ± 27.7 mm). This was immediately followed by a sustained period of below average rainfall; May's rainfall (25.3 mm) was 50% less than the mean rainfall (50.1 ± 25.8 mm) between 1880–2018 [40]. Furthermore, this was combined with an atypically hot summer, with minimum and maximum temperatures falling outside of the range expected based on historic averages [40]. Monitoring the performance of assets under a variety of environmental conditions enables us to better predict how anticipated trends and variability in climate might impact on essential services and the natural environment. Understanding this relationship will enable us to better cope with, and recover from, climate induced disruptions.

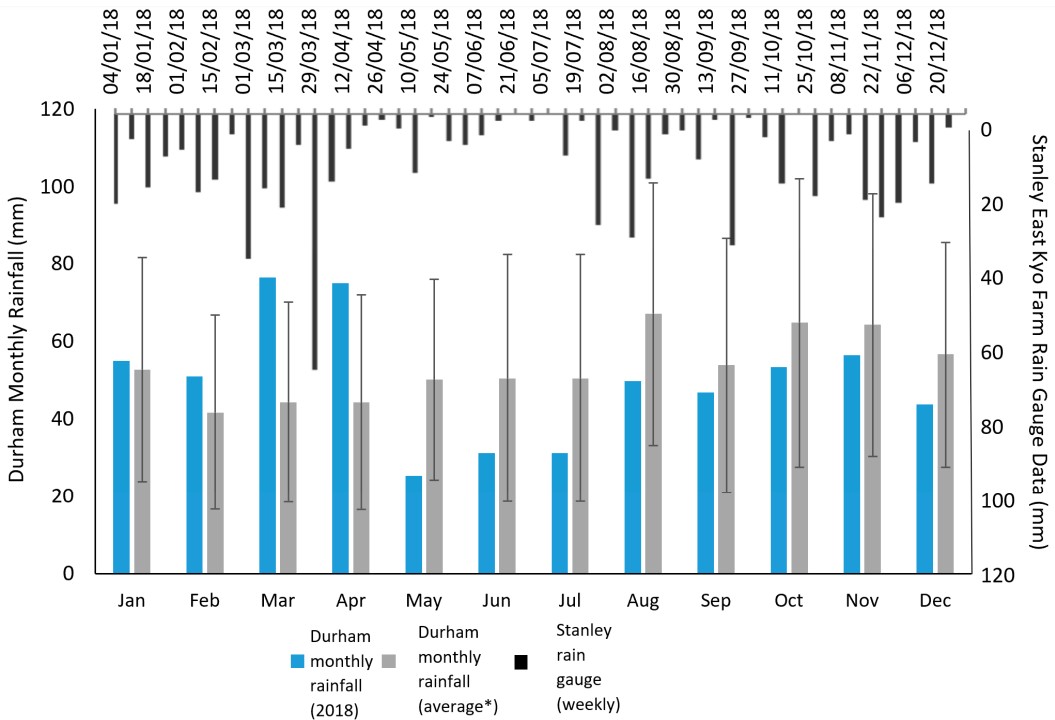

**Figure 7.** Total monthly rainfall (mm) in Durham in 2018 compared to average Met Office UK climate-historic station data (1880–2018) (primary *x*-axis) [40], and weekly total rainfall (mm) from Stanley East Kyo Farm rain gauge (secondary *x*-axis) [41]. The Stanley rain gauge is the nearest Environment Agency telemetered rain gauge to South Moor, where the data are reported from a tipping bucket rain gauge as accumulated totals for each 15 min period and summed per time interval [41].

### 3.2. Reductions in Runoff

The monitoring equipment was installed in March 2018, and data collection was planned for a 12-month period. However, there were challenges initially with the remote transfer of data, due to extremely low signal. The data collection period spans 22 March 2018–25 February 2019. Flow level was recorded at 5 min intervals. Therefore, there was the potential for 98,208 data points to be collected from each manhole during this period. There were 33,547 points recorded in manhole 1 (34.2% data completeness); 29,028 recorded in manhole 6 (29.6% data completeness); and 31,773 recorded in manhole 7 (32.4% data completeness). A rain gauge, which was installed later in the project on the community centre roof, recorded fifteen heavy rainfall events (defined by a rainfall intensity of 12–18 mm/h) between November–December 2018. However, data for the manhole level flow in this

period was not captured, perhaps due to signal strength or power issues. This period of the data collection has been omitted from the analysis below.

There were two periods where data was collected consistently, with minimal gaps, in both manhole 6 and 7: 20 August–22 October 2018; and 18 January–25 February 2019 (Figure 8). Rainfall fell within the mean historic average for these periods. During the first period there was the potential for 18,432 individual data points to be recorded for each manhole and during the second period, there was the potential for 11,232 data points. Analysis of the raw data set showed 18,155 data points (98.5% completeness) for manhole 6, and 18,284 data points (99.2% completeness) for manhole 7 over the first, 64-day period (Table 1). There were 10,856 data points (96.7% completeness) recorded for manhole 6, and 7996 data points (71.2% completeness) for manhole 7 over the second, 39-day period.

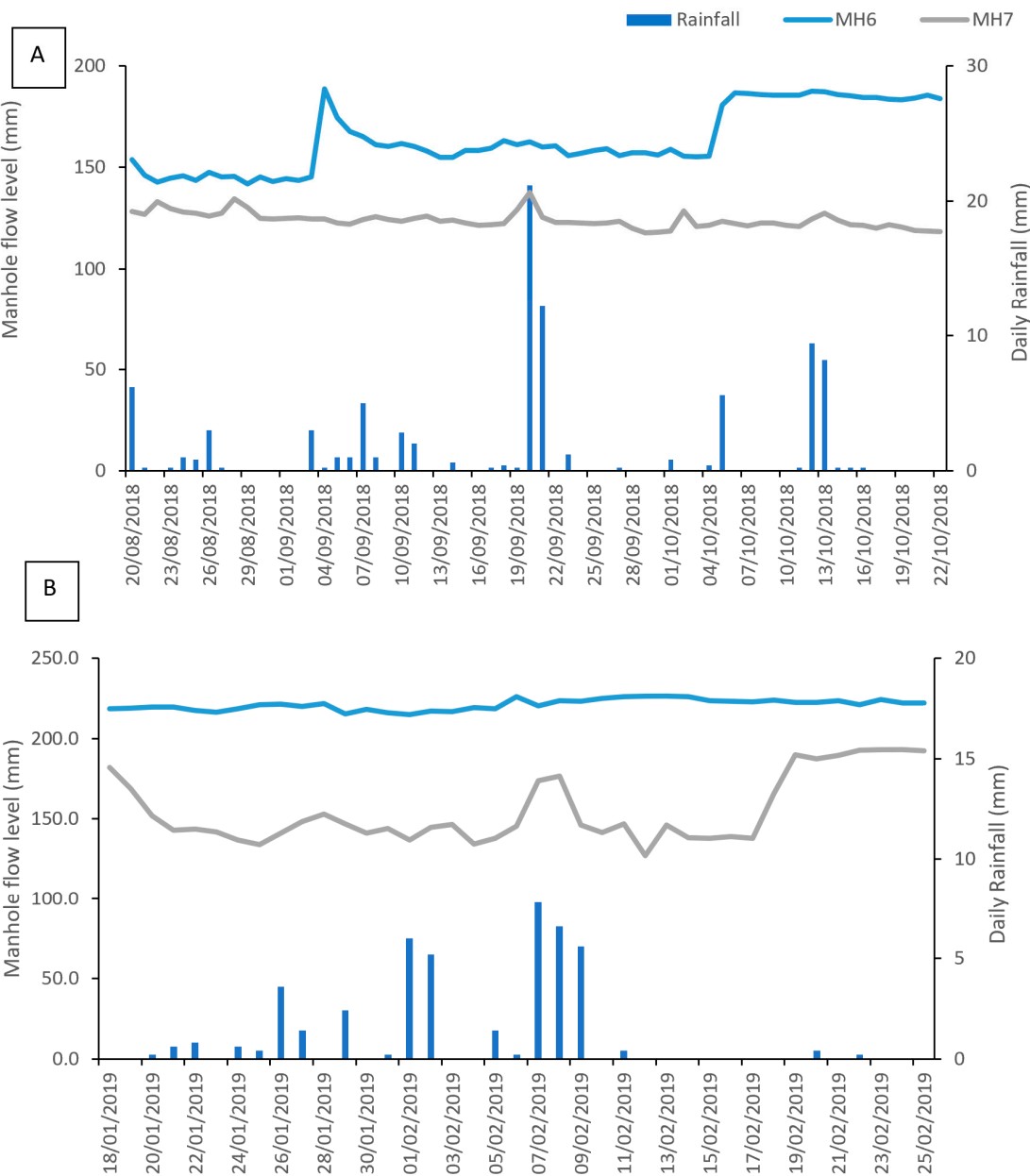

**Figure 8.** Level measurement in millimetres in manhole 6 (blue line) and manhole 7 (grey line) between 20 August–22 October 2018 (**A**) and 18 January–25 February 2019 (**B**). Daily rainfall is provided on a secondary axis in both. There were significant gaps in data collection between 23 October 2018–17 January 2019.

**Table 1.** Results of the statistical analysis of the differences in flow level (mm) recorded by the ultrasonic sensors upstream (manhole 6) and downstream (manhole 7) of the SuDS tree pits.

| Time Period | 20 August–22 October 2018 | | 18 January–25 February 2019 | |
|---|---|---|---|---|
| | **Manhole 6** | **Manhole 7** | **Manhole 6** | **Manhole 7** |
| Data completeness (%) | 98.5 | 99.2 | 96.7 | 71.2 |
| Mean flow level (mm) | 163.9 ± 15.5 | 124 ± 3.8 | 221.2 ± 3.3 | 154.8 ± 20.8 |
| Median flow level (mm) | 159.5 | 123.3 | 221.6 | 146.1 |
| Paired sample t-test | $t(63) = 18.125, p < 0.0005$ | | $t(38) = 20.114, p < 0.0005$ | |
| Related samples sign test | $z = -7.875, p < 0.0005$ | | $z = -6.085, p < 0.0005$ | |

Flow levels were averaged per day. A paired-samples t-test was run to evaluate if there was a mean difference in manhole level before and after the SuDS interventions on Poplar St (i.e., manhole 6 = "before" and manhole 7 = "after"). The level in manhole 6 was higher (163.9 ± 15.5 mm) than the level in manhole 7 (124 ± 3.8 mm) (Figure 8A). There was a statistically significant decrease in flow level in the manhole downstream of the SuDS trees compared to the flow level upstream, $t(63) = 18.125, p < 0.0005$ (Table 1). The assumption of normality of data was violated, as assessed by Shapiro–Wilk's test ($p = 0.001$) (Figure 9A,B). Whilst the paired samples t-test is robust to violations of normality, a non-parametric alternative, the Wilcoxon signed-rank test, was carried out. The distribution of differences (Figure 10A) was not symmetrical violating one of the assumptions of the test. A related-samples sign test revealed that, during first period (64 days, 20 August–22 October 2018), the median flow level in the manhole downstream of the SuDS trees was statistically significantly lower (123.3 mm) than the median flow level in the manhole upstream of these interventions (159.5 mm), $z = -7.875, p < 0.0005$ (Table 1). An ANCOVA was run to determine if there was a difference between flow level detected in two manholes either side of the SuDS features, after controlling for variation in rainfall during the monitored period. After adjusting for rainfall, there was a statistically significant difference in manhole flow level either side of the SuDS interventions, $F(1, 125) = 400.231, p < 0.0005$, partial $\eta2 = 0.762$. Data collection in manhole 1 (on the adjacent street) was not collected for the full duration of this period: only 9915 points were collected between 18th September and 22nd October 2018 (53.8% data completeness). This street did not have SuDS tree pits installed, but did have some SuDS features such as raingardens and permeable paving. The mean flow level in manhole 1 was 148.2 ± 10.4 mm for this period.

In the second period, between 18th January and 25th February, 2019 (Figure 8B) a paired-samples t-test revealed the level in manhole 6, upstream of the SuDS trees, was statistically significantly higher (221.2 ± 3.3 mm) than the level in manhole 7 (154.8 ± 20.8 mm), which was downstream of the SuDS trees, $t(38) = 20.114, p < 0.0005$ (Table 1). The assumption of normality of data was not violated for manhole 6 ($p > 0.05$) (Figure 9C), but was violated for manhole 7 ($p = 0.001$) (Figure 9D), as assessed by a Shapiro–Wilk's test. Whilst the paired samples t-test is robust to violations of normality, a non-parametric alternative (the Wilcoxon signed-rank test) was carried out. The distribution of differences (Figure 10B) was not symmetrical violating one of the Wilcoxon signed-rank assumptions. As such, a related-samples sign test showed that the median flow level in the manhole downstream of the SuDS trees was statistically significantly lower (146.1 mm) than the median flow level in the manhole upstream of these interventions (221.6 mm) during the second period (39 days, 18 January–25 February), $z = -6.085, p \leq 0.0005$ (Table 1). An ANCOVA, which controlled for variation in rainfall during the monitored period, showed there was a statistically significant difference in manhole flow level either side of the SuDS interventions, during the second monitored period $F(1, 75) = 381.315, p < 0.0005$, partial $\eta2 = 0.836$. Whilst these results are promising, they are indicative of a 25–30% reduction in flow level between the SuDS features, and not causal, as the flow measurement in manhole 7 (downstream of manhole 6) was not measured prior to the installation of the SuDS interventions. This highlights a need for gathering baseline data ahead of installing interventions to truly evaluate their impact. The gaps in the data also highlight the difficulties in deploying monitoring kit to evaluate

SuDS interventions in urban retrofit scenarios, due to both the physical environment and the way in which connections can behave unpredictably.

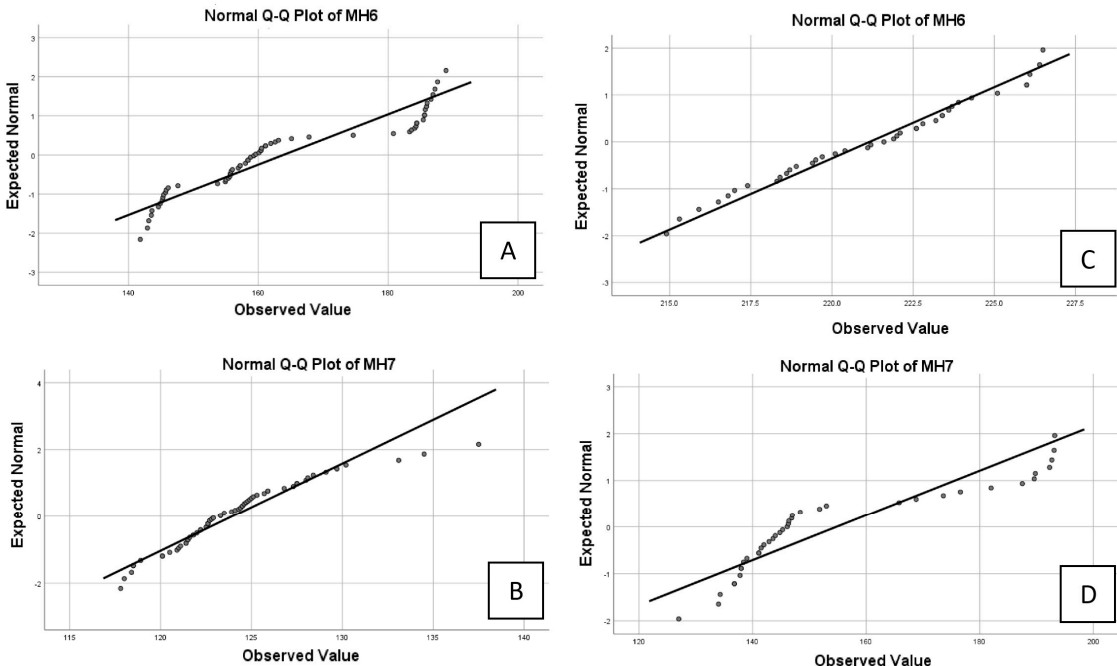

**Figure 9.** Normality plots for (**A**) manhole 6 during 20 August–22 October 2018, (**B**) manhole 7 during 20 August–22 October 2018, (**C**) manhole 6 during 18 January–25 February 2019 and (**D**) manhole 7 during 18 January–25 February 2019.

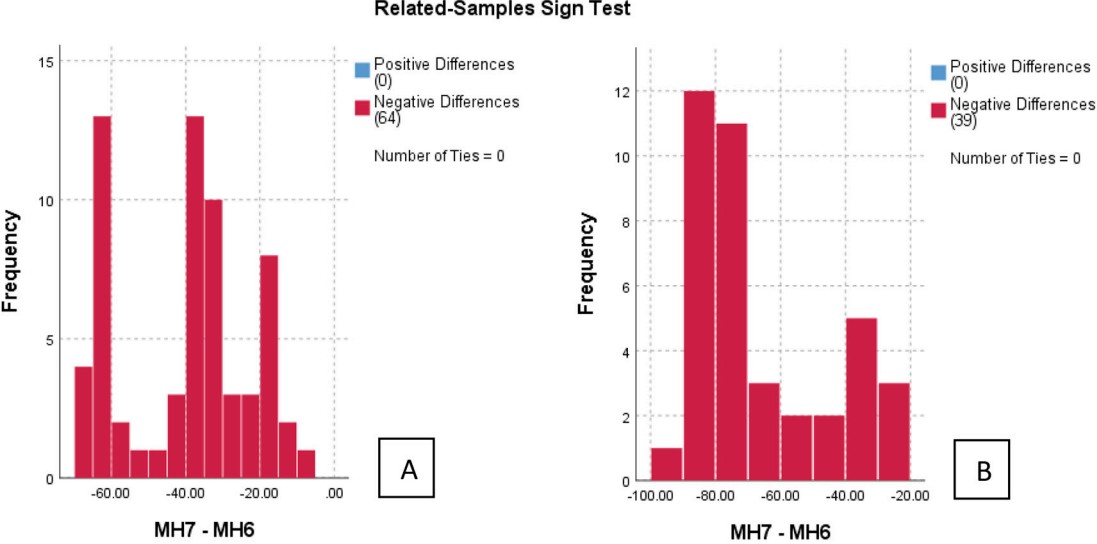

**Figure 10.** Related-samples sign test histogram showing the positive and negative differences between manhole 7 (after the SuDS trees) and manhole 6 (before the SuDS trees) during 20 August–22 October 2018 (**A**) and 18 January–25 February 2019 (**B**). Both histograms show only negative differences indicating flow level was lower downstream of the SuDS features.

## 3.3. Engagement and Adoption

Community engagement can be time-consuming to establish. However, co-designed solutions tend to fit better with community aspirations and reduce the likelihood of opposition to such schemes. This was noted by the Partnerships Manager at the Environment Agency, who reflected:

> *"Through partnership working, the demonstration site at South Moor will allow small, innovative businesses to test new water and energy saving products in a live environment. Crucially the project includes input from 300 residents to shape solutions and deliver immediate impact for the local community".*

Local residents were consulted throughout the planning stages. A community consultation reported that flooding and water quality were two of the main issues residents identified in the Twizell Burn and more than half (55.6%) wanted to see improvements, such as SuDS, to prevent or reduce the risk of flooding [35]. In this study, the aim was to involve and engage communities in the co-design of water infrastructure solutions to aid adoption, awareness and education through the use of digital and communications technologies. This presented opportunities to improve education and provide tools and resources for understanding how to protect and maintain their local environment. One of the key deliverables of the tender was to engage the community of South Moor in the surface water monitoring proposal. A range of outreach activities enabled engagement with different sectors of the community, appealing to their distinct needs and interests. For example, activities such as planting up hanging baskets and weeding the street-level raingardens were planned around informational drop-in events at the community centre, so residents could participate in hands-on activities and learn about the new water infrastructure and ongoing regeneration. This interaction with practitioners taught them new skills, which they could implement to maintain the new SuDS features or apply to their own rear-yards. It also presented the opportunity to garner informal feedback on residents' perceptions and acceptance of the new, greener streetscape. Hosting the drop-in events at the community centre presented the opportunity to showcase the live stream of data on the user interface, discuss the impact of the SuDS features and provide information on complementary topics, such as water efficiency. This aligned with further incentives to (i) improve water and energy efficiency and (ii) upskill around digital literacy. To enable this, DCC installed a computer for public use in the community centre.

Feedback suggested that despite the best intentions, community engagement with the new water infrastructure was mixed. At the street scale, the community champions were highly engaged and keen to participate in the informal drop in events and volunteering activities. A group of 5–10 residents attended all such events, contributed to maintenance tasks, engaged with the project team proactively and readily sought opportunities to increase further investment in blue-green infrastructure in the area. However, this engagement was not widespread beyond the champions, with some residents voicing concern, to the project team at drop-in events, over who was going to maintain the raingardens and prevent them becoming overgrown with weeds. Furthermore, there was some concern over the disruption caused during the installation process of the tree pits and attenuation crates (which involved a large area of the road being excavated), as well as the impact the new infrastructure had on the layout of the street, particularly in relation to large vehicle access and parking spaces.

At the property scale, uptake was also mixed. The housing regeneration manager at Durham County Council said,

> *"I think they're excellent additions to peoples' yards and it would be fantastic if we could get more of them in."*

Some residents agreed with this view, one reported,

> *"I'm very happy with it. It's nice to look at from my window, I like the plants that were planted in there, and I re-varnished the wood, because I decided I wanted something a little bit darker."*

However, another resident was less happy, reporting that they thought the planter was maintenance-free and *"didn't anticipate the work required"*, in terms of weeding and pruning. By late summer (six months after installation) the two planters were noticeably different, with one planter having substantially more plant growth. This is likely, because only one of the residents was providing extra water during the extremely dry summer.

These results suggest that stakeholder engagement in such projects is necessary in a number of ways. Firstly, engagement with the full range of stakeholders (i.e., public, EA, DCC, NWL, Durham University) was necessary to develop a relatively straightforward street scale SuDS scheme. An extra layer of complexity was added by involving monitoring, but this did not result in any other stakeholders being required. Secondly, the inclusion of the community in the project secured acceptance for the required disruption to implement the SuDS technologies. The impact on the local community necessary for retrofitting SuDS should not be underestimated. Disruption occurs to local communities through roadworks and lost parking during the installation of SuDS features, and not all of the community may like the changes to their streets. Thirdly, for SuDS to deliver their full suite of potential benefits, the communities, which are intended to benefit (in terms of health and wellbeing) needed to engage and look after their additional green space. The actions of monitoring and sustaining the new green space demand the local community take on new responsibilities that may require new skills. Assumptions should not be made that the time, skills and responsibility will be automatically given by local residents. A lack of engagement here would undermine, or potentially destroy, the sustainability and function of the implemented SuDS features.

## 4. Discussion

This study sought to demonstrate a new, collaborative process for addressing surface water management concerns. Coordinating a project with multiple drivers, stakeholders and novel technologies will inevitably present challenges. Furthermore, having to retrofit the infrastructure, rather than incorporate into the design of a new development, added to the level of project difficulty. In this section, we evaluate the effectiveness of the SuDS interventions deployed, the challenges for data capture and the social perceptions of SuDS.

### 4.1. Effectiveness of SuDS

The evidence obtained from this study suggests a mixed response to the effectiveness of SuDS. The ultrasonic sensors, installed at the street scale to quantify the volume of runoff attenuated by two SuDS tree pits, recorded a statistically significant decrease in manhole flow level downstream of the deployed interventions equating to a 25–30% reduction in mean flow level. However, the data obtained was patchy, with significant gaps arising as a consequence of signal obstacles and challenges with installing monitoring equipment in a retrofit scenario. Furthermore, any effect on the frequency of CSO spills, reductions in peak flow, or reductions in discharge rate were not evaluated and, therefore, it is difficult to ascertain the full impact of the interventions. Whilst there were challenges with quantitatively evidencing the benefits of SuDS, there was significant support for partnership working and knowledge sharing throughout the project, particularly in relation to innovative approaches and activities which could be promoted, or co-delivered, between networks and partnering organisations. The sharing of case studies such as this, is vital for capacity building and identifying learning opportunities for best practice. A study of this scale was only possible due to the collaboration and buy-in of stakeholders with adequate authority to grant access to both the physical infrastructure and the communities with whom the SuDS interventions would directly affect.

At the property scale, quantitative evidence for the performance of the rear-yard planters was not part of the study design. Therefore, no data were collected on the quantity and/or quality of water attenuated or discharged from the planters. This is a recommendation for future study. Qualitative evidence was obtained through DCC engaging with local residents and stakeholders, and findings suggest that whilst there was a high level of support for these types of interventions, communication around their maintenance and purpose could be improved. The effectiveness of future SuDS schemes is dependent on understanding both the successes, and also the challenges encountered, in deploying SuDS. Many of the difficulties encountered in monitoring the SuDS interventions illustrate some of the reasons why evaluation of SuDS is not as extensive as researchers and practitioners would like. Dissemination of studies, such as this, provide a valuable opportunity to address this.

### 4.2. Challenges for Data Capture

This study piloted a way of capturing data on the flow and attenuation of surface water in a mixed urban-rural environment. Recommendations from The Big SuDS Survey supported data collection on deployed SuDS' performance, to improve their future implementation [17]. It was hoped that data collected in this study could provide an evidence base for the performance of water infrastructure, demonstrating how that performance changed with time and with extremes of weather. However, whilst a significant difference was observed between the flow level in the manholes up and downstream of the SuDS features, the quality of information obtained was limited, due to a range of factors including experimental design and signal strength, which led to significant gaps in the data collected.

Wireless telecommunications services are relied upon for communicating and accessing information, supporting applications such as remote sensing and telemetered data. These services depend on the quality of the enabling infrastructure, but signal obstacles, such as large buildings or valleys, (or in this case, manhole covers) can result in poor coverage. Furthermore, there is evidence to support a socio-economic divide in the quality of wireless telecommunication services, with poorer connectivity experienced in rural or lower-income areas than urban or affluent areas [42].

Shortly after installation, it was noted that the telemetered unit was unable to push data to the UI in the community centre, due to extremely low signal. Several iterations of higher-gain external antennae were trialled, before opting to install a near-surface level high gain antennae to provide consistent signal for the instrumentation to transmit data. This involved cutting a small slot in the road surface and drilling into the manhole chamber to connect the telemetry unit to the road-mounted antenna, which was subsequently bonded and secured using suitable surfacing material. This solution, installed several months after the initial infrastructure, enabled the data to be transmitted and visualised in near to real time. Undertaking the installation of monitoring equipment was only possible due to the collaboration, but was time consuming. To alter physical infrastructure needed high level permission, especially to cut into the road surface. It is, therefore, very difficult to deploy and maintain monitoring equipment to routinely collect data on the effectiveness of SuDS.

### 4.3. Social Perceptions of SuDS

One of the key challenges with SuDS lies in the uncertainty around their ongoing maintenance. Community engagement with water infrastructure is therefore particularly important, because the performance of SuDS can be positively or negatively affected by the behaviours and attitudes of those that use and maintain them after their installation. Whilst the benefits of greening are widely reported, negative engagement with green infrastructure can also occur. This may arise if the local community do not have adequate time to participate in—or do not sufficiently value—community projects; if negative views of the interventions are held; or if the features translate into ecosystem "disservices", creating new concerns or issues for the community, such as crime and other risks [43]. Community interactions were only evaluated anecdotally in this project, through informal feedback provided to the project team and the local authority. Substantial delays in the installation process may have affected community engagement. Negative engagement was not thought to be a significant issue in this project, but there were isolated cases of community concern, some of which were temporary, such as the disruption caused during installation, and others which were longer-term, relating to maintenance. A repeat of the survey conducted during the community consultation period, may help to quantify public perception to the installed infrastructure. This study, therefore, supports previous findings that adoption, ownership and maintenance of SuDS remain a barrier to their uptake [17,44]. However, it highlights an opportunity to interact with communities better, and to go further to help them understand the value and impact of water infrastructure on their lives. Collaborative approaches are key to overcoming this.

The role of stakeholder engagement is, therefore, paramount to achieving public support and maintaining SuDS long-term. This includes creating awareness, and promoting understanding of SuDS purpose and function, to positively shape community and stakeholder attitudes and behaviours

to green infrastructure. However, Lamond and Everett suggest this should be considered less from a position of demographics and, instead, from an understanding of how and why different users within the community may engage with the infrastructure (i.e., what actions, activities or practice they may undertake there), rather than how frequently or when they may interact [45]. This perhaps explains why members of the community who were involved practically in the project—through drop in events and interaction with the user interface—were more supportive and willing to contribute to maintenance practices, such as weeding the raingardens, than those who were merely passing-by.

## 5. Conclusions

SuDS and NBS are being rolled out as key way of greening urban environments, reducing a range of environmental hazards and improving community wellbeing and health. This study reflects on a case study in the north east of England, highlighting both the successes of partnership working in implementing sustainable drainage interventions, and the challenges in obtaining high quality data in urban and rural areas and retrofit scenarios. Our findings suggest:

(1) Coordinating a project with multiple drivers, stakeholders and novel technologies presents challenges, but multi-institution collaboration ensures there is the resource, capacity and buy-in required to support the deployment of SuDS.

(2) There remains a need to evidence the performance and benefits of SuDS through long-term monitoring. However, the quality of wireless telecommunication services underpinning the transfer of data may limit what can be collected. Advances in novel satellite communication technologies may provide a route to obtaining high quality telemetered data in real time in rural or less-connected locations.

(3) Whilst there is a growing interest in more innovative or integrated technologies in the SuDS community, we need to ensure this translates to the local communities too. Co-creating solutions, which consider how and why local residents might engage with water infrastructure practically, will lead to technical solutions that better align with community aspirations and are more likely to be accepted and maintained in the long-term.

(4) The difficulties encountered in obtaining robust data to evidence the performance of SuDS at the street scale illustrates some of the reasons why evaluation of SuDS is not as extensive as researchers and practitioners would like.

(5) It is likely to be more straightforward to design and deliver evaluation of SuDS in new developments or non-real world test sites, but this is unlikely to deliver a holistic understanding of how SuDS work in much of our urban environments.

With the growing momentum behind the implementation of SuDS, it is crucial that we grow an informative evidence base on their effectiveness. We suggest there should be more studies to assess the physical and social impacts of the development of schemes in real world situations. Furthermore, we encourage research to monitor the performance of SuDS at a more granular level, such as the collection of data including the volume attenuated in real-time, or the quantity and quality of water discharged from SuDS features. This data would be useful to provide support for the implementation of such features individually, but also to enable the comparison, and optimisation, of SuDS technologies in a management train. Finally, we propose a number of principles to guide future studies which should: (i) use a collaborative approach across the full range of practitioners needed to implement SuDS and the community; (ii) develop novel techniques to ensure the real time monitoring of rainfall, runoff and flows can be captured from sewers; and (iii) combine investigations with a detailed mapping of surface and subsurface draining routes, to ensure monitoring is located in the most suitable points.

**Author Contributions:** S.C. was responsible for the writing—original draft preparation, data collection and analysis, and data visualisation. L.J.B. was responsible for writing—review and editing, supervision, funding acquisition and conception. All authors have read and agreed to the published version of the manuscript.

**Funding:** This research was part funded by the European Regional Development Fund (ERDF), with match funding from four project partners: Durham University, Durham County Council, Northumbrian Water Limited and the Environment Agency.

**Acknowledgments:** We thank our project collaborators Chris Jones, Niall Benson, George Gerring, Adrian Cantle-Jones, Jenny Taylor, Ashleigh Adey, Rachel Murphy, Jackie Mckenna, Nicola Bramfitt and Anna Gerring; and the businesses directly involved in this project.

**Conflicts of Interest:** The authors declare no conflict of interest.

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
