# Peer review of "Assessing the Effectiveness of Sustainable Drainage Systems (SuDS): Interventions, Impacts and Challenges"

_water, doi:10.3390/w12113160_

Round 1

Reviewer 1 Report

The authors present a very interesting paper. The introduction is robust, the methdology is well exaplined and results, and conclussions as well, are correct and interesting. I agree with the auhtors that relevant studies monitoring SUDS are lacking so I must congratulate them for dealing with this topic. Although it would be of interest to focus on larger SUDS infrastructures, they have deployed a dense monitoring network which is very valuable.
My concerns are related to three minor methodological questions. First, in my opinion the authors should go further in the explanation of the mehtodology followed for SUDS design and location. Why do they selected the specific places where SUDS facilities were located? and which criteria or standard did the follow for SUDS design? and why did they select such conceptual framework? Second, in many ocasions SUDS are aimed at reducing runoff but, reducing runoff drives to smaller discharging rate to natural water courses what can cause environmental problems to hydrologycal systems. It becomes very relevant in a region where this has been identified as a key problem. I suggest the authors to discuss this question. Finally, I also suggest the authors to discuss the consequences of SUDS installation on the existing conventional urban draining facilities in terms of flow reduction and local reuse of surface runoff.

Author Response

Reviewer: 1

The authors present a very interesting paper. The introduction is robust, the methodology is well explained and results, and conclusions as well, are correct and interesting. I agree with the authors that relevant studies monitoring SUDS are lacking so I must congratulate them for dealing with this topic. Although it would be of interest to focus on larger SUDS infrastructures, they have deployed a dense monitoring network which is very valuable. My concerns are related to three minor methodological questions.

We are pleased to hear the reviewer found the work interesting and are grateful for their comments, which we have addressed below.

First, in my opinion the authors should go further in the explanation of the methodology followed for SUDS design and location. Why do they selected the specific places where SUDS facilities were located? and which criteria or standard did the follow for SUDS design? and why did they select such conceptual framework?

We have added in more detail to section 2.2 and 2.3 about why the specific types of SuDS design were chosen, and why the study took place in this particular location. Please see lines 238-268  for more detail.  We have also added a new section (Section 2.1, line 175 -205) that introduces the conceptual framework for the research, including a new Figure 1.

Second, in many occasions SUDS are aimed at reducing runoff but, reducing runoff drives to smaller discharging rate to natural water courses what can cause environmental problems to hydrological systems. It becomes very relevant in a region where this has been identified as a key problem. I suggest the authors to discuss this question.

Unfortunately we did not monitor water quality or discharge rates to natural water courses. Therefore, we have  not included this aspect in the current paper as it goes beyond the research conducted in our existing study. We feel including such a discussion would damage the cohesiveness of the paper.

Reviewer 2 Report

The effectiveness of sustainable drainage systems is performed using t- test in this manuscript. 

  • Methodology of t-test could be mentioned in Section 2.
  • Assumption of t-test is that data are normal. Normality test results could also be demonstrated as tabular and graphical form.
  • t-test results can also be clearly demonstrated. 
  • Two groups of data are considered (MH6 and MH7). Mann-Whitney is an appropriate statistical test, if scale data are not normal. I encourage the authors to perform Mann-Whitney test. 

Author Response

Reviewer: 2

The effectiveness of sustainable drainage systems is performed using t- test in this manuscript. 

  • Methodology of t-test could be mentioned in Section 2.

The statistical methods have been added in to Section 2 – see section 2.6 (line 373-387).  

  • Assumption of t-test is that data are normal. Normality test results could also be demonstrated as tabular and graphical form.

Normality plots (Figure 9) have been added in graphical form (line 487).

  • t-test results can also be clearly demonstrated. 

Table 1 has been added to summarise statistical test results more clearly (line 460)

  • Two groups of data are considered (MH6 and MH7). Mann-Whitney is an appropriate statistical test, if scale data are not normal. I encourage the authors to perform Mann-Whitney test. 

We thank the reviewer for their suggestion. Unfortunately, the Mann Whitney U is a non-parametric alternative of the independent t-test. As we conducted a paired-samples t-test, a Mann Whitney U would not be appropriate. We repeated the analysis with a Wilcoxon signed-rank test which is a non-parametric alternative of the paired t-test. However, the distribution of differences was not symmetrically shaped, and therefore, one of the assumptions was violated and this test could not be used. In this instance, the related-samples sign test  can be used to compare differences instead as it does not have an assumption of a symmetrically shaped distribution of differences. The results from this test can be seen on lines 443-449 and lines 469-474.

Reviewer 3 Report

This study was very interesting and I enjoyed reading it.  The paper was well thought out and I appreciate how honest the authors were regarding the problems with data collection.  I have just a few points

Line 316 - 318 needs to be removed

Line 360 - 387 Could you put the results of the statistical significance tests in a table form for readability.

Line 397-492 Could you discuss in greater detail how you evaluated community interactions and the potential wellbeing benefits of the SuDS intervention.

Author Response

Reviewer: 3

This study was very interesting and I enjoyed reading it.  The paper was well thought out and I appreciate how honest the authors were regarding the problems with data collection.  I have just a few points

We’re pleased to hear the reviewer enjoyed reading the manuscript and appreciated our honesty around the challenges we had faced.

Line 316 - 318 needs to be removed

Apologies. This has been removed.

Line 360 - 387 Could you put the results of the statistical significance tests in a table form for readability.

Table 1 has been added to summarise statistical test results more clearly (line 460).

Line 397-492 Could you discuss in greater detail how you evaluated community interactions and the potential wellbeing benefits of the SuDS intervention.

More detail has been added to section 3.3 on engagement and adoption – please see lines 508-538 and lines 505-535 and in the discussion section 4.3 on social perception of SuDS – please see lines 640-646.

Round 2

Reviewer 2 Report

I am OK with the revisions provided by the authors. This manuscript now could be published in the journal.

Thanks for considering me in the reviewing process of this manuscript.